# Age-Associated Changes of Sirtuin 2 Expression in CNS and the Periphery

**DOI:** 10.3390/biology12121476

**Published:** 2023-11-29

**Authors:** Maider Garmendia-Berges, Noemi Sola-Sevilla, MCarmen Mera-Delgado, Elena Puerta

**Affiliations:** 1Pharmaceutical Sciences Department, Division of Pharmacology, School of Pharmacy and Nutrition, University of Navarra, 31008 Pamplona, Spain; mgarmendia.5@alumni.unav.es (M.G.-B.); nsola.4@alumni.unav.es (N.S.-S.); mmeradelgad@alumni.unav.es (M.M.-D.); 2Navarra Institute for Health Research (IdiSNA), 31008 Pamplona, Spain

**Keywords:** aging, brain, epigenetics, inflammation, neurodegenerative diseases, sirtuin 2

## Abstract

**Simple Summary:**

In the last decades there has been a demographic growth of the elderly population, a trend that will continue due to the increase in life expectancy. Hence, understanding the molecular mechanisms underlying aging and identifying ways to modulate those mechanisms are intriguing areas of research. In this context, several studies have shown age-related changes in sirtuin 2 protein in humans and in different animal models, suggesting that it has a central role in aging and age-related pathologies. In the brain, most studies seem to demonstrate an increase in sirtuin 2 expression during the aging process, supporting that it may represent an ideal new target for the treatment of age-related neurodegenerative diseases. In contrast, in peripheral tissues, sirtuin 2 levels appear to decrease with aging; thus, systemic administration of any known modulator of this protein would have conflicting outcomes. This review summarizes the currently available information on changes in sirtuin 2 expression in aging, with the aim of providing an up-to-date overview of the topic and understanding its potential as a pharmacological target to treat age-related diseases.

**Abstract:**

Sirtuin 2 (SIRT2), one of the seven members of the sirtuin family, has emerged as a potential regulator of aging and age-related pathologies since several studies have demonstrated that it shows age-related changes in humans and different animal models. A detailed analysis of the relevant works published to date addressing this topic shows that the changes that occur in SIRT2 with aging seem to be opposite in the brain and in the periphery. On the one hand, aging induces an increase in SIRT2 levels in the brain, which supports the notion that its pharmacological inhibition is beneficial in different neurodegenerative diseases. However, on the other hand, in the periphery, SIRT2 levels are reduced with aging while keeping its expression is protective against age-related peripheral inflammation, insulin resistance, and cardiovascular diseases. Thus, systemic administration of any known modulator of this enzyme would have conflicting outcomes. This review summarizes the currently available information on changes in SIRT2 expression in aging and the underlying mechanisms affected, with the aim of providing evidence to determine whether its pharmacological modulation could be an effective and safe pharmacological strategy for the treatment of age-related diseases.

## 1. SIRT2 and Aging

In the last decades, with the increase of life expectancy, there has been a demographic growth in the elderly population, a tendency that is expected to continue. According to the World Health Organization, it is expected that from 2015 to 2050 the world’s population over 60 years old will almost double, raising from 12% to 22% [1]. As the tendency to disease increases with age, the focus on healthy aging and its research is becoming more relevant. Considering the aging process is very diverse, if we aim to understand it, the influence of genetics should not be ignored. Gene expression can be altered in many levels such as DNA replication, transcription, RNA translation, and post-translational modifications of proteins. In this context, epigenetics could also be a piece of the puzzle. Epigenetics is defined as “the study of changes in gene function that are mitotically and/or meiotically heritable and that do not entail a change in DNA sequence” [2]. These changes occur through several mechanisms, all of which can be modulated by environmental, physiological, and pathological processes. Some of the predominant epigenetic mechanisms are ATP-dependent chromatin-remodeling complexes, non-coding RNAs, covalent modifications of DNA bases and histone modifications [3].

In particular, histone modifications have been widely studied. Histones are octameric proteins around which DNA strands wrap, conforming the chromatin. These DNA-binding proteins provide a structural support to DNA, that can be extended or compacted depending on the modifications (methylation, acetylation, phosphorylation, and ubiquitination) that take place in their tails. Notably, histone acetylation on the lysine residues reduces chromatin condensation becoming more accessible to transcription factors, and activating the transcription of genes located in the affected region. This results in an increase gene expression. Histone acetyltransferases (HAT) and histone deacetylases (HDAC) are crucial in these epigenetic modifications since they are responsible for dynamic histone acetylation and deacetylation, respectively, tuning the level of transcripts (for a review, see [4]) (Figure 1).

Eighteen HDAC proteins are classified into four classes named I–IV. Class III HDACs, also known as sirtuins, are nicotinamide adenine nucleotide- (NAD^+^) dependent, while the other three classes require zinc as a cofactor. Sirtuins also differ from other HDACs in their unique ability to catalyze ADP-ribosylation [5,6].

Sirtuins are present in every living species and widely distributed along the tissues [7]. Besides deacetylating histone proteins, they are also key in the control of many physiological processes in mammals. For instance, they take part in the regulation of the cell cycle, antioxidant protection, inflammation, neurogenesis, and various metabolic pathways. Thus, their functions are often related to stress response, aging, and general homeostasis [8,9].

To date, seven isoforms of sirtuins (SIRT1-7) have been described in mammals. They all differ in their functions among other reasons due to diverse terminal regions, subcellular location, enzymatic activities, and substrates [10,11]. Regarding their subcellular location, SIRT1 is mostly located in the nucleus, with the ability to migrate to the cytosol. SIRT2, on the other hand, is the only sirtuin that is predominantly cytosolic, although it has the capacity to shuttle to the nucleus and mitochondria. SIRT3, SIRT4, and SIRT5 are mitochondrial sirtuins, being SIRT3 and SIRT5 able to migrate to the cytoplasm. SIRT6 is found in the nucleus, associated with chromatin, and SIRT7 in the nucleolus and nucleus [9,12,13,14].

Although their functions are very diverse and many studies are still needed to understand their role in each process, it has been described that SIRT1 is involved in inflammation, oxidative stress, cell proliferation, and apoptosis [15,16], while the other nuclear sirtuins, SIRT6 and SIRT7, participate in DNA repair [17] and regulation of gene transcription [18], respectively. Mitochondrial sirtuins SIRT3, SIRT4, and SIRT5 are involved in the response to oxidative stress and metabolic pathways inside this organelle [19]. SIRT2 has been involved in multiple functions regulating gene expression and many metabolic pathways [20].

Since the interest in sirtuins is growing, research is being carried out on all the members of the sirtuin family in different experimental models, tissues, and conditions, leading to new findings that intend to clarify their physiological and pathological relevance. Specifically, several recent studies have described age-related changes in SIRT2 in different organs and tissues [21,22,23,24]. This has led to the hypothesis that SIRT2 could play a key role in the aging process. However, these changes are not the same in all tissues, since an increase [25,26,27] or a decrease [28,29,30] in its expression has been described depending on the tissue analyzed. Therefore, a detailed understanding of the function of SIRT2 with aging in each cell type is necessary to determine if it is an interesting target for the treatment of diseases associated with aging. In this context, the present review will try to summarize and organize the main studies that address the role of SIRT2 in aging. 

The human *SIRT2* gene is located on chromosome 19q13.2 and its protein is formed by 389 amino acids [8,20]. This protein is highly distributed along the organism; for instance, it is expressed in the liver, heart, kidney, skeletal muscle, adipose tissue, macrophages, and brain [7,20].

The list of substrates deacetylated by SIRT2 is extensive (for a review see [12]) explaining its implication in a wide variety of functions. In the cytoplasm, one of its main known functions is cytoskeleton stabilization by the deacetylation of α-tubulin [31]. In addition, during mitosis, SIRT2 deacetylates histone 4 at lysine 16 and changes chromatin structure from a transcriptionally active to a repressive state [32]. It collaborates in the regulation of the cell cycle since there is a rise in its nuclear expression during the G2/M transition [12], leading to the hypothesis that it could be a crucial player in the development of tumors (reviewed in [33,34]). Moreover, it has been corroborated that SIRT2 participates in DNA replication, transcription, and RNA translation by deacetylating crucial substrates such as ATR kinase checkpoint [35], RNR [36], TIF-IB/SL1 [37], and eIF5A [38]. Thus, it promotes the replication, the binding of transcription factors, and the initiation of translation (for a review, see [20]).

Other SIRT2 substrates include proteins involved in cellular metabolism like LDH-A, GAPDH, ENO1, and ALDOA [39,40]. Moreover, it has been suggested that SIRT2 reduces ROS levels and therefore protects against oxidative stress via deacetylation of FOXO1, FOXO3a, G6PD, PGAM2, and NF-kB [41]. Noteworthy, Zhang and coworkers, by a quantitative proteomic analysis of the lysine acetylome, have recently identified new SIRT2 substrates [42] which undoubtedly will lead to increasing knowledge of SIRT2 and the functions in which it is involved.

Interestingly, SIRT2 is often associated with aging and longevity and it has been suggested to be an important modulator of age-related pathologies. It has been pointed out that some polymorphisms of *SIRT2* result in reduced lifespan [43]. Specifically, carriers of the SIRT2-rs45592833-T allele showed a decreased chance of surviving longer. In fact, subjects with a copy of minor allele T have a reduced probability of about 3.7 times to become long-lived people compared to the most frequent allele G, while this probability diminishes 13 times for subjects with two copies of allele T. This association between *SIRT2* polymorphism and longevity was sex-independent. Further in silico analysis identified three different miRNAs (miR-3170, miR-92a-1-5p, and miR-615-5p) which tend to bind more tightly to the T allele, resulting in a reduction of SIRT2 expression [43]. This study suggests that low levels of SIRT2 may have a negative impact on aging and longevity.

However, on the other hand, in vitro studies using U2OS cells treated with doxorubicin, (an inducer of premature senescence) showed elevated *Sirt2* mRNA and protein levels. This indicates that the upregulation of SIRT2 is specifically linked to stress-induced premature senescence and thus, SIRT2 levels could be used as an indicator of cellular senescence [22].

In this context, a sizable number of studies have found age-related changes of SIRT2 [25,44] in humans and different animal models; however, they reach apparently contradictory conclusions on whether it increases or decreases with age, whether these changes are beneficial or detrimental and the exact relationship is unknown. A detailed analysis of the relevant works published to date shows us that the changes that occur in SIRT2 with aging seem to be inverse in the central nervous system (CNS) and in the periphery, a fact that is reviewed and discussed in the following sections.

## 2. CNS SIRT2 Expression in Aging

SIRT2 is the most abundant member of the sirtuin family in the CNS, notably in the cortex, striatum, hippocampus, and spinal cord [12]. In recent years, this fact has exponentially increased interest in SIRT2 in neuroscience research in order to decipher its implication in aging and age-related neurodegenerative disorders.

The first study on this topic was published by Maxwell and colleagues in 2011 [25]. In this study, the authors found that three isoforms predicted for SIRT2 (SIRT2.1, SIRT2.2, and SIRT2.3) are expressed in the brain. They analyzed young adult (4–5-month-old) and aged (19–22-month-old) C57BL/6 mice and found an age-related accumulation of the isoform SIRT2.3 in spinal cord extracts and cortices. As a result, total SIRT2 levels had a modest but significant increase in aged mice. Notably, they detected an association between the local accumulation of SIRT2 protein and areas of reduced tubulin acetylation in cell bodies and neurites which could be affecting neuronal function. More recently, the behavioral and molecular consequences of the overexpression of SIRT2.3 in the hippocampus have been addressed [45]. According to this work, the overexpression of SIRT2.3 does not result in relevant behavioral or molecular changes in control mice. However, in a mouse model of accelerated aging, the SAMP8 model, SIRT2.3 overexpression worsened learning and memory performance and increased the expression of the pro-inflammatory cytokine IL-1β. Based on these results, the increase of SIRT2.3 in aged brains does not seem to induce or prevent senescence, but it could play a part in the progression of age-related processes together with other risk factors. 

In line with Maxwell’s findings, a study checked for SIRT2 level variations in various brain areas of female Wistar rats at 3, 12, and 24 months old. An increase in *Sirt2* mRNA and protein levels was observed, but exclusively in the occipital lobe. This increase was paired with a significant enhancement in deacetylated FOXO3a (Forkhead Box, class O3a) transcription factor, a substrate deacetylated by SIRT2, in the same region [24]. Supporting this data, a recent study, analyzing the cortex and hippocampus of 3- and 22-month-old Wistar albino male rats, has shown an increase in SIRT2 and FOXO3a brain levels during the aging process, accompanied by an increase in oxidative stress and apoptosis [27]. In addition, authors randomly administered melatonin, which is reported to have antioxidant, anti-apoptotic, and anti-aging properties, and is physiologically reduced in aging; the SIRT2 inhibitor salermide, or DMSO as a control to both young and aged rats. They described a reduction in SIRT2 and FOXO3 protein levels in the hippocampus but not in the cortex of aged rats treated with melatonin. Interestingly, salermide administration to aged rats led to the inhibition of SIRT2 and FOXO3 in both regions. Considering these results and the functions attributed to melatonin in aging, the authors suggest that SIRT2 and FOXO3 could play a key role in oxidative stress and apoptosis [27]. Indeed, the FOXO transcription factors are regulated by post-translational modifications, and SIRT2-mediated deacetylation of FOXO3a promotes its ubiquitination and degradation [46]. In fact, SIRT2 deacetylates FOXO3a and stimulates its translocation to the nucleus, therefore inducing apoptosis [47]. In addition, exposure to oxidative stress upregulates FOXO3a in the hippocampus [48], enhanced levels of SIRT2 lead to cell death, and the inhibition of SIRT2 results in a reduction of oxidative stress and apoptosis [22,49,50,51,52]. In agreement with these observations, an increase in FOXO3a activity has been found in neurodegenerative diseases such as Alzheimer’s disease and Parkinson’s disease [53,54]. However, the reason why the increase in SIRT2 is only observed in certain brain regions and the physiological consequences of these changes need to be further investigated.

Supporting the notion that elevated SIRT2 levels in the CNS are deleterious, another study has shown an upregulation of SIRT2 in the brain of a D-galactose-induced aging rat model [55]. In fact, as a result of D-galactose administration, the expression of the pro-inflammatory cytokines interleukin-6 (IL-6), and tumor necrosis factor-alpha (TNF-α) increased, while the autophagic marker Beclin-1 was downregulated. Noteworthy, metformin supplementation induced an anti-aging effect, downregulating the expressions of SIRT2, IL-6, and TNF-α, whereas increasing Beclin-1 expression. The authors state that metformin promotes the activation of autophagy and reduces inflammation, hence restoring the antioxidant status and improving brain aging [55]. In line with these results, the implication of SIRT2 in autophagy has also been described in different studies. It has been shown that upregulated SIRT2 interferes with autophagy efficiency and promotes protein accumulation under proteasome inhibition, intensifying proteinopathy-related cytotoxicity [56]. Consistently, lowered SIRT2 increases autophagy levels [57]. Taking into account that, in the context of neurodegenerative disorders, inadequate autophagy induces neuronal cell death while activated autophagy is neuroprotective, these results further justify the deleterious consequences of age-related increases in SIRT2 in the CNS.

In a different publication, Diaz-Perdigon and colleagues [26] compared SIRT2 protein and mRNA levels in 2- and 9-month-old male SAMR1 and SAMP8 mice. In both models, they found a significant increase in hippocampal SIRT2 protein in aged animals, with no significant differences between both strains. Therefore, they pointed out SIRT2 as a possible biomarker of the aging process. However, this increase did not correlate with changes in *Sirt2* mRNA, which according to the authors, indicates protein accumulation rather than an increase in its synthesis. In contrast, there were no significant changes in the protein expression in the frontal cortex and striatum. Interestingly, in order to understand the physiological consequences of the observed SIRT2 increase, they administered the SIRT2 inhibitor 33i to 5- and 8-month-old SAMP8 and SAMR1 mice. Authors conclude that early SIRT2 inhibition improves age-related cognitive decline and prevents neuroinflammation in SAMP8 mice. However, the inhibition of SIRT2 once the aging phenotype is well established (in 8-month-old SAMP8 mice) cannot reverse age-induced behavioral and molecular changes [26]. These results point to SIRT2 inhibition as a promising therapeutic target to prevent age-related cognitive decline.

In agreement with all these studies, it has been recently published that SIRT2 protein expression increases gradually with aging in the cortex and hippocampus isolated from 3-, 6-, 12-, and 24-month-old C57BL/6 wild type (WT) mice [58]. Interestingly, the authors show that, at the same time, SIRT1 expression decreases gradually; thus, the SIRT2:SIRT1 ratio gradually increases with age. In an attempt to understand how the changes in SIRT1 and SIRT2 levels may affect the vulnerability of the neurons to a neurotoxic insult, SH-SY5Y neuroblastoma cells were transfected with empty vector, flag-tagged SIRT1 or SIRT2, and then treated with Aβ42 oligomers. They found that Aβ substantially increased cell death when transfecting cells with an empty vector, whereas SIRT1 overexpression largely restored the cell damage by Aβ. On the other hand, SIRT2 overexpression reduced the survival of Aβ42-treated cells compared to untreated cells [58]. Together, these data support the notion that SIRT1 and SIRT2 have inverse effects on neuron viability; SIRT1 protects against neurotoxicity, while SIRT2 promotes it.

The findings of the studies mentioned above point out an increase of SIRT2 in the CNS during the aging process. However, another study performed by Kireev and colleagues showed differing results when testing male Wistar rats [59]. In this case, the researchers found a significant age-related decrease of *Sirt2* mRNA accompanied by an increase in gene and protein levels of pro-apoptotic markers (Bax and Bad) in the dentate gyrus comparing 2- and 22-month-old animals [59]. Noteworthy, in this case, growth hormone treatment reduced the pro/anti-apoptotic ratio to levels observed in young animals and also increased SIRT2 levels, which was accompanied by a reduction in apoptosis and enhanced survival markers in this part of the hippocampus.

In general terms, most of the studies collected in this section agree in concluding that SIRT2 seems to be increased in the CNS with aging (Table 1) and that this increase seems to be harmful by promoting oxidative stress and neurodegeneration. Therefore, based on these conclusions, SIRT2 inhibition or different strategies aimed at counteracting age-related increases in SIRT2 could be considered good therapeutic options for age-related diseases.

## 3. Peripheral SIRT2 Expression in Aging

Considering that the expression of SIRT2 is very extensive in the periphery, several works have also addressed the changes that occur in its expression in different peripheral cell types (Table 2), leading to opposite conclusions to those reached when SIRT2 was analyzed in the CNS.

In 2007, Chambers and colleagues [21] carried out a study using highly purified bone marrow hematopoietic stem cells (HSC) from 2- and 21-month-old C57BL/6 mice. They analyzed the expression of more than 14,000 genes identifying 1500 that were age-induced and 1600 that were age-repressed. As expected, the up-regulated genes were associated with the stress response, inflammation, and protein aggregation, whereas the down-regulated group was marked by genes involved in maintaining genomic integrity and chromatin remodeling. Among them, SIRT2 was found to be significantly reduced with aging [21]. In their conclusions, authors highlight the epigenetic perspective of aging, which elucidates the diversity of the effects of age at the molecular, cellular, and organ levels. In agreement with this study, a few years later, Luo et al. [29] also studied HSCs isolated from bone marrow of 3-month-old and 24-month-old C57BL/6 mice. Again, they found a reduction in *Sirt2* mRNA levels in old mice compared to the young ones [29]. Aiming to elucidate whether these changes are cause or consequence of aging and to assess the functions of SIRT2 in HSCs, they analyzed HSCs in WT versus SIRT2 knockout (KO) mice. They observed that old SIRT2 KO mice had fewer HSCs in bone marrow compared to old WT mice. In addition, they found a decrease in lymphoid and an increase in myeloid cells in the peripheral blood of aged SIRT2 KO mice, which implies that SIRT2 has an age-dependent effect on HSC maintenance and hematopoiesis. They also demonstrate that SIRT2 promotes HSC survival upon the activation of the NLRP3 inflammasome, and suggest, for the first time, that SIRT2 could modify the NLRP3 activity at the post-transcriptional level. Thus, reduced SIRT2 expression in aged HSCs explains the age-induced upregulation of the NLRP3 inflammasome. Supporting this hypothesis, NLRP3 downregulation or SIRT2 overexpression counteracted the functional decline of HSC with aging [29].

More recently, He and colleagues [44] measured *Sirt2* mRNA levels in macrophages isolated from bone marrow of male C57BL/6 mice, and they found that the expression in this cell type was reduced in 24-month-old compared to 3-month-old animals. In agreement with the aforementioned observations, they demonstrate that SIRT2 deacetylates NLRP3 leading to the inactivation of the NLRP3 inflammasome. This serves as evidence supporting the physiological significance of the acetylation switch in the NLRP3 inflammasome, thereby regulating inflammation associated with aging and influencing glucose homeostasis. In addition, they demonstrate that 2-year-old SIRT2 KO mice fed a chow diet exhibit metabolic alterations, insulin resistance, and peripheral chronic inflammation. These results indicate that insulin sensitivity maintenance and repression of NLRP3 inflammasome activation during aging necessitate SIRT2 [44]. This research reveals a mechanism of inflammaging and points out that aging-associated conditions can be reversed by upregulating SIRT2 or promoting NLRP3 deacetylation.

Supporting these studies, another publication has also reported an inverse correlation between SIRT2 levels and age in peripheral blood mononuclear cells (PBMC) of healthy humans [60]. The authors point out the involvement of SIRT2 in aging biology and suggest that it may be a potential biomarker for monitoring health conditions and aging [60]. However, in another study, the analysis of SIRT2 expression in the peripheral blood of healthy adults (25–35 years old) and elderly people (65 years old and over) led to the opposite conclusion: a significant age-related increase in *SIRT2* mRNA levels was found [28].

Due to the high prevalence of cardiovascular diseases in aging, several recent studies have also focused on studying whether SIRT2 could be playing an important role in the correct functioning of this system. Ye et al. [30] have compared hearts from aged (18- to 21-year-old) and young (4- to 6-year-old) cynomolgus macaques. They found that the size of the aged monkey’s cardiomyocytes doubled the size of the younger ones. They also revealed an increased cardiac fibrosis and staining of senescence-associated β-galactosidase with age. In order to determine the underlying molecular mechanisms of these phenotypic differences, they carried out mass spectrometry-based proteomics in cardiomyocytes. As expected, they found upregulation of proteins involved in pro-inflammatory response, blood clotting, and fibrosis; and downregulation of proteins in pathways related to protein synthesis, mitochondrial function, and lipid metabolism, in aged versus young hearts. Among them, they observed, for the first time, a sex-independent and age-related decrease of the SIRT2 protein in heart samples. Interestingly, the cardioprotective role of SIRT2 was supported by in vivo experiments in mouse models. Intramyocardial injection of lentiviruses expressing SIRT2 resulted in an improved cardiac dysfunction in aging [30]. This study supports the notion that SIRT2 is a key protein in the periphery; specifically, it exhibits cardioprotective effects and can be proposed as a potential therapeutic target against age-related cardiomyocyte hypertrophy and associated cardiac dysfunction.

In agreement with these conclusions, another recent study has also proposed that SIRT2 may serve as a potential therapeutic target for vascular rejuvenation [61]. The study describes that among the sirtuin family, SIRT2 is the most abundant in human and mouse aortas, an expression which is reduced with aging. Interestingly, old SIRT2 KO mice show accelerated vascular aging (arterial stiffness and constriction–relaxation dysfunction, accompanied by aortic remodeling, collagen deposition, and inflammation), which correlates with mitochondrial oxidative stress and transcriptome reprogramming. Moreover, they also show that SIRT2 is also relevant for aging and related vascular diseases in humans. Using a public proteome dataset researchers found that plasmatic SIRT2 is decreased with aging and is a valuable predictor of age-related aortic diseases in humans [23].

Supporting the beneficial consequences of maintaining peripheral SIRT2 expression in different organs in aging, it has been demonstrated that SIRT2 could have a key role in the mechanisms underlying caloric restriction. Caloric restriction, without malnutrition, is the most effective and reproducible physiological intervention promoting longevity from yeast to mammals (reviewed in [62]). It is the most consistent non-genetic and non-pharmacological approach to extend lifespan, acting through a reduction in insulin and insulin-like growth factor, and an increase of insulin sensitivity [55]. In particular, SIRT2 expression increased in the kidney and white adipose tissue of mice in response to caloric restriction [47]. It is hypothesized that some of the beneficial consequences of caloric restriction could be mediated by this increase in SIRT2 since it deacetylates BubR1, a promoter of a healthy lifespan that is physiologically reduced in aging. It has been described that mice overexpressing BubR1 live longer than the hypomorphic ones, which in addition show signs of accelerated aging. Noteworthy, by keeping lysine-668 of BubR1 deacetylated, SIRT2 promotes BubR1 stability therefore increasing the lifespan of BubR1 deficient mice [63].

Together, these data coincide in demonstrating that SIRT2 decreases in the periphery with age and highlight the importance of maintaining its expression to delay aging and age-related diseases and further investigating the molecular mechanisms underlying this beneficial effect.

## 4. Is SIRT2 a Good Pharmacological Target for Age-Related Neurodegenerative Diseases?

Confirming the hypothesis that the increase in SIRT2 observed in the CNS with aging is harmful, numerous studies have also shown that its expression is also increased in different neurodegenerative diseases and that its genetic deletion or pharmacological inhibition provides beneficial effects in these conditions (Figure 2).

### 4.1. Alzheimer’s Disease

Among all age-related neurodegenerative diseases, Alzheimer’s disease (AD) is the most common form of dementia with over 50 million people suffering worldwide. This disease is characterized by a progressive cognitive decline and memory impairment accompanied by other neuropsychiatric symptoms including apathy, anxiety, sleep disturbances, and depression [64]. AD brain is characterized by the presence of extracellular senile plaques of aggregated amyloid-β (Aβ) peptide, and intracellular aggregations of hyperphosphorylated Tau, the neurofibrillary tangles. Both hallmarks together with a chronic neuroinflammatory reaction lead to a progressive neurodegeneration. Current efforts to find a cure for AD have been unsatisfactory and the drugs currently available to treat this disease only address the cognitive and behavioral symptoms, with limited effectiveness. 

In this context, several studies have shown increased expression of SIRT2 in AD brains [65,66,67,68]. Confirming these results, SIRT2 has been identified as a potential cerebrospinal fluid biomarker discriminating AD from other neurological diseases [69]. These results support the association of SIRT2 expression in the CNS and AD pathology and the potential therapeutic interest of SIRT2 inhibition in this disease [70]. In fact, it has been recently demonstrated in different AD mouse models that SIRT2 genetic deletion or pharmacological inhibition improves learning and memory, restores alterations in long-term potentiation, and reduces amyloid and tau pathology and neuroinflammation [65,66,71,72,73]. Interestingly, it has been demonstrated that SIRT2-mediated deacetylation of the amyloid precursor protein (APP) promotes the amyloidogenic processing and therefore Aβ formation and accumulation [73]. Accordingly, SIRT2 genetic deletion or inhibition in the APP/PS1 model, increased APP acetylation, reduced Aβ formation, improved memory, and reduced mortality [73]. Additional mechanisms contributing to the reduction in amyloid pathology observed after SIRT2 pharmacological inhibition include the inhibition of β-secretase activity [72] and the increase in Aβ engulfment by microglia [65]. All these studies support SIRT2 inhibition as an effective therapeutic strategy in the treatment of AD.

### 4.2. Parkinson’s Disease

Parkinson’s disease (PD) is an age-associated neurodegenerative disorder characterized by the progressive loss of dopamine-producing neurons from the substantia nigra pars compacta, which causes deficiencies in dopamine levels in the brain. In addition, these brains present aggregates of α-synuclein called Lewy bodies [74]. These changes lead to primarily motor symptoms, but olfactory disorders, cognitive decline, and depression are also present in this type of patient [75].

Regarding SIRT2, an increase in its expression levels has been described in samples of PD mouse models [76,77,78] and PD patients [67,79]. In addition, high enzymatic activity of SIRT2 has been found in postmortem brain tissue from patients with PD compared to elderly controls [67] and two polymorphisms (rs2015 and rs2241703) in the 3′-UTR of the *SIRT2* gene have been associated with PD risk. All this evidence suggests that SIRT2 could be playing an important role in this disease. In this sense, several studies have demonstrated that SIRT2 deletion or inhibition showed neuroprotective effects in rotenone and MPTP-induced animal models of PD [76,78,80,81,82]. Moreover, it has been recently demonstrated that α-synuclein is a substrate of the deacetylase activity of SIRT2. Interestingly, SIRT2 mediated-deacetylation of α-synuclein promotes its aggregation and neurotoxicity. Accordingly, SIRT2 inhibition or genetic deletion rescued α-synuclein toxicity and showed neuroprotective effects in different in vitro and in vivo models of PD [80,83,84,85,86,87], demonstrating the potential therapeutic value of SIRT2 inhibition in synucleinopathies and, more specifically, in PD.

### 4.3. Huntington’s Disease

Huntington’s disease (HD) is a progressive, fatal hereditary autosomal dominant neurodegenerative disorder with multiple neurological manifestations. The genetic basis of HD is a CAG trinucleotide repeat (40 or more times) expansion within exon 1 of the huntingtin gene (HTT). The mutant HTT protein is prone to misfolding and aggregation inside the cells leading to cell death. The neuropathological changes are found in the cortex and striatum, and patients suffer from motor dysfunction, dementia, emotional disturbances, and premature death (for review, see [88,89]).

Regarding SIRT2, it has been shown that mRNA levels of SIRT2 are increased in the striatum of postmortem HD brains [89], supporting its potential interest as a pharmacological target in this disease. In this sense, neuroprotective effects of pharmacological and/or genetic inhibition of SIRT2 have been demonstrated in different models of HD [90,91,92,93]. Luthi-Carter and colleagues [90] suggested that these neuroprotective effects could be due to the transcriptional repression of cholesterol biosynthesis. The implication of SIRT2 in modulating cholesterol biosynthesis was corroborated later by Taylor et al. [94]. Moreover, Chopra and coworkers [91] observed that SIRT2 inhibition with AK-7 compound reduced HTT aggregates, improved motor function, reduced brain atrophy, and extended survival of two genetic mouse models of HD. Similar neuroprotective effects have been described in the SIRT2 knockdown Drosophila model of HD [92,93].

Nevertheless, some contradictory results have also emerged. Bobrowska et al. [95] showed that genetic reduction or ablation of SIRT2 did not modify the disease progression or HTT levels in the HD genetic mouse model R6/2. The reason for these discrepancies is unclear; thus, more studies are needed to confirm the specific effects of SIRT2 on HD.

### 4.4. Amyotrophic Lateral Sclerosis

Amyotrophic lateral sclerosis (ALS) is a progressive, incurable, and fatal neurodegenerative disorder, characterized by the loss of upper and lower motor neurons that control voluntary muscle movement. With a lifetime risk of 1 in 300 people, the mortality rate is high with most deaths caused by respiratory failure due to the loss of voluntary muscle control [96]. Clinically, it is a highly heterogeneous disease with different sites of symptom onset, different clinical manifestations (upper motor neuron versus lower motor neuron degeneration versus cognitive symptoms), and different progression. The underlying mechanisms driving this phenotypic diversity remain unknown.

Unlike other neurodegenerative diseases described above, SIRT2 does not appear to play a fundamental role in the development of this disease. It has been described that *Sirt2* mRNA expression is increased in the spinal cord in different transgenic mouse models of ALS including G93A-SOD1 and G86R-SOD1 mice; however, protein expression remains unchanged in all the models examined [97]. Moreover, SIRT2 genetic deletion did not modify the progression of the disease and failed to provide any beneficial effect in the SOD1G93A mouse [98]. These results rule out the potential interest in using pharmacological SIRT2 inhibitors for the treatment of ALS.

Interestingly, more recently, it has been shown that SIRT2 expression is specifically upregulated in cognitively unaffected ALS patients, proposing that its elevated expression could be a marker of cognitive resilience in ALS pathogenesis [99]. Among the underlying mechanisms of this beneficial effect mediated by SIRT2, authors show that increases in SIRT2 were associated with decreases in NLRP3 inflammasome activation. Accordingly, they demonstrated a significant upregulation of NLRP3 in cognitively affected ALS individuals. This study supports previous research suggesting that inappropriate activation of the inflammasome plays a role in neurodegenerative diseases, including ALS [100,101]. From this perspective, and taking into account that SIRT2 is an inhibitor of NLRP3 activity [44], pharmacological inhibition of SIRT2 would be harmful and aggravate the development of these diseases. Thus, although most of the studies reviewed in this section support the interest of SIRT2 inhibition in different neurodegenerative diseases, further investigations are needed to fully understand the underlying mechanisms and signaling pathways involved before the translation of these results to human patients. 

## 5. SIRT2 Pharmacological Inhibitors

The growing interest aroused by the therapeutic potential of SIRT2 has promoted the design of new molecules that inhibit its enzymatic activity. However, despite their promising therapeutic results in preclinical studies, none have been approved due to their low selectivity for SIRT2, potency, or physicochemical properties.

Regarding the neuroscience field, where SIRT2 inhibition seems to be an efficient pharmacologic strategy, AK-1, AK-7, and AGK-2 are the most used inhibitors in cellular and in vivo models of neurodegenerative diseases. In this sense, even though AK-1 (IC50 = 12.5 µM) is more potent than AK-7 (IC50 = 15.5 µM), it lacks blood–brain barrier (BBB) permeability, a crucial characteristic for the treatment of neurodegenerative diseases [102]. In addition, AGK-2 is even more potent than AK-1 and 7 (IC50 = 3.5 µM), but it does not cross the BBB either [103]. Another compound called 33i (a 2-anilinobenzamide derivative) (IC50 of 0.57 µM) exhibits potent and selective SIRT2 inhibition in enzyme assays compared to previous reported SIRT2 inhibitors with more than 10-fold greater SIRT2-selectivity over SIRT1 and SIRT3 compared to AGK-2 [103]. Moreover, it crosses the BBB [26,65,104] and has been shown to provide beneficial effects in different AD mouse models [26,65]. Other compounds called 17k [105], ICL-SIRT078 [84], and γ-mangostin [106] have been also designed in the last few years.

In addition to these molecules, other strategies have also been used to inhibit SIRT2 activity, such as peptides (YKK(ε-thioAc)AM) [107] or microRNAs (miR-212-5p) [78]. Moreover, a new machine-learning-based tool has been recently developed by Djokovic and colleagues [108]. This tool, called *SIRT2i_Predictor*, is able to predict which molecules could be selective and potent SIRT2 inhibitors, prioritizing the best compounds and reducing the time and cost of developing novel inhibitors [108].

The development of new pharmacological inhibitors of SIRT2, more specific and with better physicochemical properties, will allow a better understanding of the functions modulated by this enzyme, a fundamental step for a possible translation to the clinic in those pathologies in which the effectiveness/risk balance of this treatment has been favorable. 

## 6. Conclusions

The studies analyzed in this review show diverse results and reach different conclusions when trying to understand the role of SIRT2 on aging. Nevertheless, these results do not need to be contradictory, since they evidence how SIRT2 expression could vary depending on the context, experimental models, methods, and samples analyzed. However, a relationship between aging and SIRT2 levels is undoubtedly exposed. 

Most of the studies explained in the CNS section seem to agree in demonstrating an increase in the expression of SIRT2 in the brain with aging (see Table 1). Although there are some differences regarding the areas analyzed and there is no clear consensus on whether this increase is harmful [27,55] or a beneficial compensatory mechanism [67], most of these results suggest that the increase in SIRT2 levels could be playing a detrimental role in aging. Thus, in this scenario, its pharmacological inhibition is proposed as a neuroprotective strategy against aging and neurodegenerative diseases associated with it. Indeed, SIRT2 pharmacological inhibition has been proven to be effective in different models of HD [91,93], PD [81,83,84,85], and AD [65,71,72,73]. However, in our opinion, these results should be analyzed critically before any clinical attempt. This consideration arises from the following observations:

Firstly, in contrast to that observed in the CNS, SIRT2 levels seem to be physiologically reduced with aging in the periphery (see Table 2), and its overexpression would be protective in the context of inflammation, vascular health, and cardiovascular diseases. These differences between the effects of SIRT2 at the central and peripheral levels should not be overlooked when suggesting therapies that inhibit SIRT2 since systemic inhibition could have severe consequences in peripheral organs and systems that are also crucial in aging. In this context, the results of a work recently published by our team clearly align with the findings exposed in this review [65]. We administered a SIRT2 inhibitor to APP/PS1 mice, a model for AD, and found beneficial effects in the CNS: improved memory and learning, decreased amyloid pathology, and reduced neuroinflammation. However, when we assessed the consequences of the inhibition in the periphery, we found a relevant increase in inflammation. In this sense, the role of SIRT2 might be differential throughout the organism and therefore its inhibition should be cautious.

Secondly, the pattern of expression of SIRT2 is not even conserved between the different brain regions; thus, systemic administration of any known modulator of this enzyme would have conflicting outcomes. In this sense, validation of brain-region-specific transcriptomics data necessitates the employment of complementary, spatially resolved techniques to confirm cell-type-specific differences between groups. Indeed, spatial transcriptomic approaches should be used to further investigate age-related changes in SIRT2 at the cellular level.

Finally, current knowledge of SIRT2 is incomplete. It is still unknown all the different functions and substrates modulated by this enzyme in each cell type and condition, which implies that SIRT2 modulation may have different consequences depending on the cell type affected, the disease, or the time-point the treatment is administered. This knowledge is essential to establish exactly under which circumstances, in which moment, and in which cell type the inhibition or overexpression of SIRT2 will be an effective and safe therapeutic strategy for the treatment of each disease. At that time, the next challenge will be to design a therapy specifically directed to the target cell type, avoiding the rest of the cells to maximize the therapeutic potential while minimizing adverse effects.

Therefore, although it is clear that SIRT2 has an interesting role in the aging process and could be considered a strategic pharmacological target to prevent and/or reverse diseases associated with it, further investigations are still needed before proposing the translation of this strategy to human patients. 

## Figures and Tables

**Figure 1 biology-12-01476-f001:**
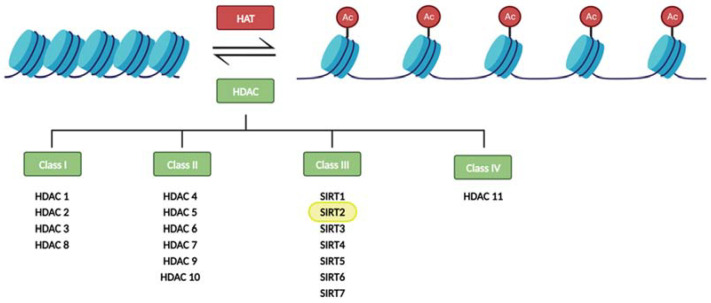
Regulation of histone acetylation levels by histone acetyl transferases (HATs) and histone deacetylases (HDACs). Sirtuins constitute the class III of HDACs.

**Figure 2 biology-12-01476-f002:**
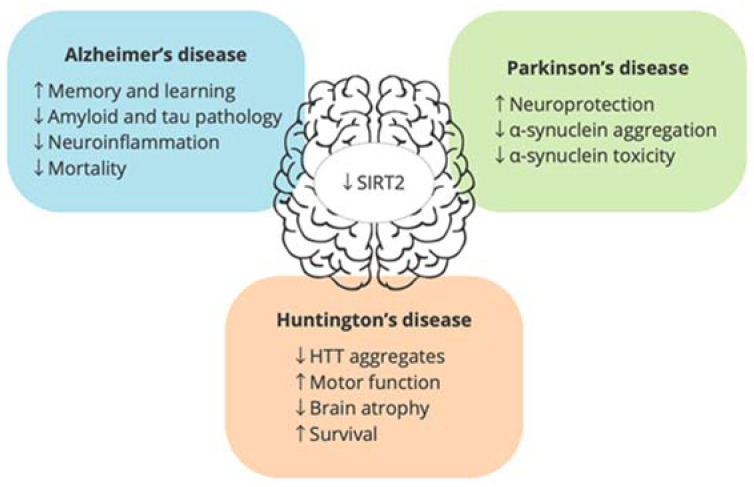
Beneficial effects of SIRT2 inhibition or deletion in preclinical models of neurodegenerative diseases. ↑ and ↓ refer to an increase and decrease, respectively, in the parameter that follows.

**Table 1 biology-12-01476-t001:** Sirtuin 2 expression variations in the aging CNS.

Authors and Year	Analyzed Model		Sample	Sirtuin 2 Expression with Aging
	Specie	Ages Compared in Months		
Maxwell et al., 2011 [25]	C57BL/6 mouse	4–5 vs. 19–22	Spinal cord and cortex	Increase
Kireev et al., 2013 [59]	Male Wistar rat	2 vs. 22	Hippocampus (dentate gyrus)	Decrease
Braidy et al., 2015 [24]	Female Wistar rat	3 vs. 12 vs. 24	Occipital lobe	Increase
Garg et al., 2017 [55]	Male Wistar rat	4 vs. 24	Whole brain	Increase
Diaz-Perdigon et al., 2020 [26]	Male SAMR1 and SAMP8 mice	2 vs. 9	Hippocampus	Increase
Keskin-Atkan et al., 2022 [27]	Male Wistar rat	3 vs. 22	Hippocampus and cortex	Increase
Li et al., 2023 [58]	C57BL/6 mouse	3 vs. 6 vs. 12 vs. 24	Hippocampus and cortex	Increase

**Table 2 biology-12-01476-t002:** Sirtuin 2 expression changes with aging in the periphery.

Authors and Year	Analyzed Model		Sample	Sirtuin 2 Expression with Aging
	Specie	Ages Compared		
Chambers et al., 2007 [21]	C57BL/6 mouse	2- vs. 21-month-old	HSCs isolated from BM	Decrease
Yudoh et al., 2015 [60]	Human	22- to 66-year-old	PBMCs	Decrease
Luo et al., 2019 [29]	C57BL/6 mouse	3- vs. 24-month-old	HSCs isolated from BM	Decrease
Wongchitrat et al., 2019 [28]	Human	25- to 35-year-old vs. ≥65-year-old	Peripheral blood (plasma)	Increase
Lehallier et al., 2019 [23]	Human	18- to 95-year-old	Peripheral blood (plasma)	Decrease
He et al., 2020 [44]	Male C57BL/6 mouse	3- vs. 24-month-old	Macrophages isolated from BM	Decrease
Ye et al., 2023 [30]	Cynomolgus macaque	4- to 6- vs. 18- to 21-year-old	Cardiomyocytes	Decrease
Zhang et al., 2023 [61]	C57BL/6 mouse	4- vs. 24-month-old	Aorta and VSMCs	Decrease

BM: bone marrow; HSC: hematopoietic stem cell; PBMC: peripheral blood mononuclear cell; VSMC: vascular smooth muscle cell.

## Data Availability

No new data were created or analyzed in this study. Data sharing is not applicable to this article.

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
