# Peer review of "Age-Associated Changes of Sirtuin 2 Expression in CNS and the Periphery"

_biology, 2023, doi:10.3390/biology12121476_

Round 1
Reviewer 1 Report
Comments and Suggestions for Authors
The review article, "Age-associated Changes of Sirtuin 2 Expression in CNS and the Periphery," offers a comprehensive exploration of SIRT2's involvement in various conditions, including aging and neurodegenerative disorders. While the article successfully delves into these topics, it is crucial to provide a more focused analysis of SIRT2's role in neurodegenerative conditions. This can be achieved by incorporating a separate subheading dedicated to elucidating SIRT2's impact on neurodegenerative diseases, thereby enhancing the reader's understanding of this critical aspect. Under this dedicated subheading, the article should intricately detail the multifaceted roles of SIRT2 in different neurodegenerative disorders such as Alzheimer's disease, Parkinson's disease, and amyotrophic lateral sclerosis. Discussing its involvement in disease aetiology, progression, and potential therapeutic implications will provide readers with a comprehensive view of SIRT2's significance in the context of neurodegeneration.
Reviewer 2 Report
Comments and Suggestions for Authors
This review summarizes the current knowledge on the differential expression of Sirtuin 2 (SIRT2) in the central nervous system (CNS) and periphery during aging. While the manuscript presents an interesting issue, there are some problems in the present version that need to be addressed through substantial revision.
The following is a list of comments and concerns with the manuscript:
1. The abstract suggests that this manuscript is a systematic review (“A detailed analysis of all the works published to date”), however, this is not evident in the manuscript.
2. In the introduction, the author should emphasize SIRT2's role during aging and aging-related diseases, and how its differential expression could impact aging in different tissues. This will help to capture readers' interest and introduce the topic effectively.
3. The sections on "Aging and epigenetics" and "The sirtuin family" have been extensively reviewed in many previous works and do not provide new knowledge or concepts relevant to the theme. It is recommended to combine these sections with "SIRT2 and aging" and summarize the latest research progress.
4. Abbreviations should be explained consistently throughout the manuscript.
5. In the sections "CNS SIRT2 expression on aging" and "Peripheral SIRT2 expression on aging" the authors primarily reference existing work without providing concrete explanations. As this is a review article, the authors should not simply cite what is already in the literature, but should also present the accomplishments, describe the possible pathways involved in detail, provide their own insights, suggest possible directions, highlight challenges, and offer more than just a compilation of existing information.
6. The manuscript should discuss the impact of age-associated changes in SIRT2 expression on aging-related diseases.
7. There are too few figures in the main part of the article. Additional figures would enhance the understanding of the topic.
8. The conclusion should focus on the existing problems and future directions in this field, rather than solely considering a pharmacological approach.
Comments on the Quality of English LanguageModerate editing of English language required.
Reviewer 3 Report
Comments and Suggestions for Authors
In this manuscript, the authors have summarized the changes in the expression levels of Sirtuin2 (SIRT2) during the aging process in both the brain and the periphery. In various brain regions, most studies seem to demonstrate an increase in SIRT2 expression during the aging process. In contrast, in the peripheral tissues, SIRT2 levels appear to decrease with aging. SIRT2 may represent an ideal new target associated with age-related cognitive decline and neurodegenerative changes. It is recommended that the authors consider revising the simple summary and abstract sections to improve clarity and emphasize the key points.
In fact, there is no clear contradiction in the expression changes of sirtuin 2 related to age in the central nervous system and periphery in various studies. In various brain regions, most studies seem to demonstrate an increase in SIRT2 expression during the aging process. In peripheral tissues, studies also almost unanimously suggest that SIRT2 levels appear to decrease with aging. Therefore, I believe that this article should delve more deeply into the discussion of the potential of SIRT2 in treating age-related diseases, including the regulation of SIRT2 as a drug target and its clinical significance, as well as the complex roles of SIRT2 in neurological diseases. The second important point is to summarize the existing SIRT2-specific small molecule modulators and their potential as therapeutic drugs. So I hope the author can make the necessary revisions before publishing.
Comments on the Quality of English LanguageThe manuscript appears to have a few grammar issues. I recommend that the authors thoroughly review the entire manuscript to address these issues and enhance the overall English style for greater clarity and coherence.
Round 2
Reviewer 2 Report
Comments and Suggestions for Authors
The authors addressed all of my comments.